SOFTWARE

# MGSurvE: A framework to optimize trap placement for genetic surveillance of mosquito populations

**Héctor M. Sánchez C.**[1]*, **David L. Smith**[2,3], **John M. Marshall**[1]

**1** Divisions of Epidemiology and Biostatistics, University of California Berkeley, Berkeley, California, United States of America, **2** Institute for Health Metrics and Evaluation, University of Washington, Seattle, Washington, United States of America, **3** Department of Health Metrics Sciences, School of Medicine, University of Washington, Seattle, Washington, United States of America

* sanchez.hmsc@berkeley.edu

## Abstract

Genetic surveillance of mosquito populations is becoming increasingly relevant as genetics-based mosquito control strategies advance from laboratory to field testing. Especially applicable are mosquito gene drive projects, the potential scale of which leads monitoring to be a significant cost driver. For these projects, monitoring will be required to detect unintended spread of gene drive mosquitoes beyond field sites, and the emergence of alternative alleles, such as drive-resistant alleles or non-functional effector genes, within intervention sites. This entails the need to distribute mosquito traps efficiently such that an allele of interest is detected as quickly as possible—ideally when remediation is still viable. Additionally, insecticide-based tools such as bednets are compromised by insecticide-resistance alleles for which there is also a need to detect as quickly as possible. To this end, we present MGSurvE (Mosquito Gene SurveillancE): a computational framework that optimizes trap placement for genetic surveillance of mosquito populations such that the time to detection of an allele of interest is minimized. A key strength of MGSurvE is that it allows important biological features of mosquitoes and the landscapes they inhabit to be accounted for, namely: i) resources required by mosquitoes (e.g., food sources and aquatic breeding sites) can be explicitly distributed through a landscape, ii) movement of mosquitoes may depend on their sex, the current state of their gonotrophic cycle (if female) and resource attractiveness, and iii) traps may differ in their attractiveness profile. Example MGSurvE analyses are presented to demonstrate optimal trap placement for: i) an *Aedes aegypti* population in a suburban landscape in Queensland, Australia, and ii) an *Anopheles gambiae* population on the island of São Tomé, São Tomé and Príncipe. Further documentation and use examples are provided in project's documentation. MGSurvE is intended as a resource for both field and computational researchers interested in mosquito gene surveillance.

**Data Availability Statement:** There are no primary data in the paper; all materials are available at https://github.com/Chipdelmal/MGSurvE and we have archived our code on Zenodo https://zenodo.org/doi/10.5281/zenodo.8084252. Visualizations of

several use case examples are also available on our YouTube playlist at https://youtube.com/playlist?list=PLRzY6w7pvIWrOSwOIu_MXbOr14wx9xuwT.

**Funding:** HMSC and JMM were supported by funds from the Bill & Melinda Gates Foundation (INV-017683) and the UC Irvine Malaria Initiative awarded to JMM. JMM was supported by a National Institutes of Health R01 Grant (1R01AI143698-01A1) awarded to JMM. DLS was supported by funds from the Bill and Melinda Gates Foundation (INV 030600), and National Institute of Allergies and Infectious Diseases (R01 AI163398), awarded to DLS. The funders had no role in the study design, data collection and analysis, decision to publish, or preparation of the manuscript.

**Competing interests:** The authors have declared that no competing interests exist.

## Author summary

Mosquito-borne diseases such as malaria and dengue fever continue to pose a major health burden throughout much of the world. The impact of currently-available tools, such as insecticides and antimalarial drugs, is stagnating, and gene drive-modified mosquitoes are considered a novel tool that could contribute to continuing reductions in disease transmission. Gene drive approaches are unique in the field of vector control in that they involve transgenes that could potentially spread on a wide scale, and consequently, surveillance is expected to be a major cost driver for the technology. This is needed to monitor for unintended spread of intact drive alleles, and the emergence of alternative alleles such as homing-resistance alleles and non-functional effector genes. Additionally, surveillance of insecticide-resistance alleles is of interest to support the impact of insecticide-based tools such as bednets. Here, we present MGSurvE, a computational framework that optimizes trap placement for genetic surveillance of mosquito populations in order to minimize the time to detection for an allele of interest. MGSurvE has been tailored to various features of mosquito ecology, and is intended as a resource for researchers to optimize the efficiency of limited surveillance resources.

## Introduction

Mosquito-borne diseases such as malaria, dengue and yellow fever continue to pose a major public health burden throughout much of the world. Gene drive-modified mosquitoes have been proposed as a potentially transformative tool to complement currently-available tools by biasing inheritance in favor of an introduced allele [1]. Progress has been made in *Anopheles* malaria vectors towards two general classes of gene drive strategies: i) "population replacement", whereby inheritance is biased in favor of an allele that confers refractoriness to pathogen transmission [2, 3]; and ii) "population suppression", whereby vector populations are suppressed by biasing inheritance in favor of an allele that induces a severe fitness cost or sex bias [4, 5]. In *Aedes* arboviral vectors, a "split drive" system has been engineered with Cas and guide RNA (gRNA) drive components at separate loci [6]. Split drive systems display transient drive behavior because the Cas and gRNA components frequently co-occur following a release, but soon dissociate and are eliminated by virtue of fitness costs. The potential spread and scale of impact of this technology is promising; however, surveillance programs present a major cost driver as they must scale with the intervention [7, 8].

Surveillance for gene drive projects will be required to monitor the effectiveness of the strategy at field sites, as has been done for previous self-limiting genetic control projects [9]; however, a more demanding task will be to detect unintended spread of gene drive alleles beyond field sites, and to detect the emergence of alternative alleles both within and beyond field sites. One concern for these systems is the emergence of drive-resistant alleles which, especially for population suppression strategies, would have a significant fitness advantage over intact drive alleles, leading vector populations to rebound [10, 11]. This is also a concern for population replacement strategies, as when the drive has spread and there are fewer cleavable wild-type alleles remaining, less costly resistant alleles may replace the drive alleles, reducing their duration of impact [12]. Another concern for population replacement strategies is the emergence and spread of drive alleles lacking a functional effector gene [13].

For spread beyond field sites, open questions relate to the optimal density and placement of traps and the frequency of sampling required to detect gene drive alleles, drive-resistant alleles or non-functional effector genes while they can still be effectively managed. Similar questions

relate to the spread of new alleles conferring resistance to other interventions, such as bednets or indoor residual spraying with insecticides. Lessons may be learned from studies of invasive species, a key result for which is that early detection is critical to minimizing invasion impact, preserving the possibility of local elimination, and maximizing the cost-effectiveness of surveillance programs [14, 15].

To this end, we present MGSurvE: an analytical framework that optimizes trap placement for surveillance of mosquito populations such that the time to detection of an allele of interest is minimized. MGSurvE takes into account biological features of mosquitoes and the landscapes they inhabit—namely, resources required by mosquitoes (e.g., blood and sugar-based food sources and aquatic breeding sites) and movement of mosquitoes between these resources on a landscape. It also accommodates traps with differing attractiveness profiles. MGSurvE may be used in parallel with MGDrivE [16] or MGDrivE 2 [17] to determine the expected distribution of times to detection or the number of individuals having the allele of interest at this time point in closed populations. We describe how to set up, run and interpret output from MGSurvE, and provide examples of trap placement optimization for an *Aedes aegypti* population in Queensland, Australia, and an *Anopheles gambiae* population on the tropical island of São Tomé, São Tomé and Príncipe. We then conclude with a discussion of future modeling needs and applications for genetic surveillance of mosquito populations.

## Design and implementation

MGSurvE provides a computational framework to distribute mosquito traps through a landscape such that the time to detection of an allele of interest is minimized. To do so, the MGSurvE package includes three major components (Fig 1), described here: i) "landscape specification," in which mosquito sites (or groups of sites) are attributed to nodes, with movement rates between nodes determined by movement rules and dispersal kernels, ii) "trap optimization," in which the spatial distribution of a given number of mosquito traps is optimized by minimizing the expected time for an allele of interest to reach a trap, as determined by an optimization routine, and iii) "analysis and visualization of results," in which optimization reports are exported, and landscapes including traps may be visualized These components are reflected in the structure of the codebase, which is developed in Python [18]. We now describe the model components here, with the mathematical representation provided in the S1 Text.

### Landscape specification

Before the distribution of traps can be optimized, a landscape must first be specified. In MGSurvE, the landscape is a metapopulation within which discrete mosquito population nodes are distributed. The appropriate scale at which populations are modeled depends on the species of interest and the resolution at which optimized trap placement is desired. For instance, a household scale may be adequate for *Ae. aegypti* populations, which are thought to be relatively local dispersers usually found within 50 m of the breeding site they hatched from [19, 20], while a village scale may be more appropriate for *An. gambiae* populations, which disperse over distances of up to 7 km [21]. In this initial version of MGSurvE, all sites are assumed to have equal population size. Nodes in MGSurvE may also represent specific resources—e.g., blood and sugar sources for feeding, and water sources for egg-laying—the inclusion of which allows traps to be distributed in relation to these. Landscapes in MGSurvE may be sex-specific, which is particularly relevant if specific resources are included, as only females blood-feed and lay eggs, while both females and males sugar-feed.

Once a point set of mosquito population nodes has been defined, the next step is to define the daily per-capita movement probabilities between each pair of nodes. Movement is assumed

# A. Landscape Definition

1. Load pointset
2. Define mosquito movement kernel
3. Define traps and their attractiveness kernels
4. Instantiate the landscape with provided information

# B. Optimization

1. Select type of optimization (discrete or continuous)
2. Select GA operators (mutation, crossover, selection)
3. Select or define fitness function
4. Run the optimization process

# C. Analysis and Verification

1. Export results to disk
2. Plot traps' optimized positions
3. Check GA's performance
4. Change parameters and re-run the optimization process, if necessary

**Fig 1. Components of the MGSurvE framework.** The MGSurvE package includes three major components, reflected in the codebase, to distribute mosquito traps through a landscape such that the time to detection of an allele of interest is minimized: (A) landscape specification, (B) optimization of trap distribution, and (C) analysis and visualization of results.

to be Markovian (i.e., the conditional distribution of future states depends only on the current state), and is calculated from dispersal kernels, which derive relative daily movement probabilities from the distance between each pair of coordinates and the density of other sites in the vicinity of the focal site. The base MGSurvE implementation provides functions to implement exponential decay, long-tailed exponential and zero-inflated exponential kernels; which encompass a basic family of short-distance flight-types for mosquitoes. The zero-inflated exponential, for example, takes into account the *Aedes*' tendency to dwell in a given point only to fly to nearby locations; whereas *Anopheles* disperse further from their immediate neighborhood, which can be characterized by a decaying-exponential kernel. However, the MGSurvE framework is not limited to these, and any function that takes two node coordinates and the required parameters can be defined and used. In the event that specific resources are included in a landscape, movement probabilities are modified by a "masking matrix," which is determined by the resource type of the mosquito's current node. For instance, a mosquito currently in a node with blood-feeding resources may be more likely to seek a node with water for egg-laying. This type of movement is similar to that of the MBITES framework (Mosquito Bout-based and Individual-based Transmission Ecology Simulator) [22].

With a point set and movement probabilities defined, the final step in specifying the landscape is to define and incorporate traps, the positions of which will be updated through the iterations of the optimization process. MGSurvE can incorporate a wide range of traps (e.g., BG Sentinel traps, CDC light traps, ovitraps, etc.), which may differ in their attractiveness as defined by parameters and attributes such as mean radius of attractiveness, mosquito sex, and resource

type of the current mosquito node. For instance, an ovitrap will be most likely to attract nearby female mosquitoes that are currently in a node with a food resource and hence may soon seek an egg-laying site. Of note, specific traps can be flagged as "immovable," so the positions of other traps may be optimized given the fixed trap locations. The trap attractiveness profiles are used to modify the movement probabilities of the movement matrix, and it is this modified matrix that is used by the optimization algorithm to optimize trap placement. A demonstration of landscape specification on a six-node metapopulation with resource types is provided in Fig 2.

## Optimization of trap distribution

MGSurvE makes use of a genetic algorithm (GA) to optimize trap placement by calling the DEAP framework (Distributed Evolutionary Algorithms in Python) [23], by default. GAs are a subset of evolutionary algorithms that search computational solution space using biologically-inspired operators such as mutation, crossover and selection upon computationally-created "chromosomes," which store information about potential solutions to the optimization task at hand [24]. Notably, these computational constructs not related to the biological application at hand, but rather named after the biological counterparts they emulate. In the case of MGSurvE, chromosomes consist of a list of "alleles", each of which contains information pertaining to the position of a trap. MGSurvE considers two classes of optimization problems: i) "discrete optimization", in which traps may only be placed within the set of currently-listed population nodes, and ii) "continuous optimization", in which traps may be placed anywhere in the landscape. Discrete optimization may be appropriate for applications on a larger spatial scale—e.g., for cases where population nodes are villages and traps may only be placed within villages. Continuous optimization may be appropriate for applications on a finer scale—e.g., where nodes represent specific resources (blood, sugar or water sources), and traps are placed relative to these. For discrete optimization, the chromosomes of the GA consist of a list of trap IDs (identification numbers), while for continuous optimization, the chromosomes consist of a list of coordinates representing trap longitude and latitude.

In each generation of the optimization cycle, the GA calculates the fitness of all the computational chromosomes in the current population (as defined in Equation 8 in S1 Text), which gives us an estimate of how good each potential solution is. With this information, the algorithm selects chromosomes to populate the next generation of the algorithm (selecting the ones with higher fitness more frequently than others). At this point, the GA selects pairs of chromosomes and combines them together in a process that emulates mating, copying the offspring to the new solution population. Finally, some of these newly generated solutions are selected for a process that resembles mutation, whereby some alleles of the chromosome are changed in hopes that these changes might lead to fitter individuals. By iterating these steps on our chromosomes, the population as a whole should move towards increasingly better (fitter) solutions. A thorough explanation of how the GA and how it connects to the mathematical components used in MGSurvE, along with descriptions of the terminology, is available in S1 Text. Of note, our optimization task attempts to reduce time to detection so the fitness metric of interest is being minimized; but all the principles remain the same regardless.

## Analysis and visualization of results

MGSurvE provides a number of functions to analyze and visualize results from the trap placement optimization procedure. Landscape and optimization reports are generated and exported to disk for performance checks and further analysis. Optimal trap placement can be plotted alongside the distribution of mosquito population nodes, with examples provided in the following use case section and Figs 2–4. Plots from MGSurvE are fully compatible with matplotlib

## A. Homogeneous site types

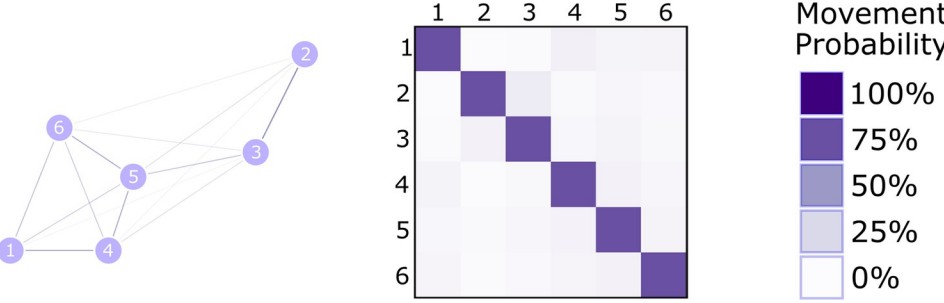

## B. Heterogeneous site types

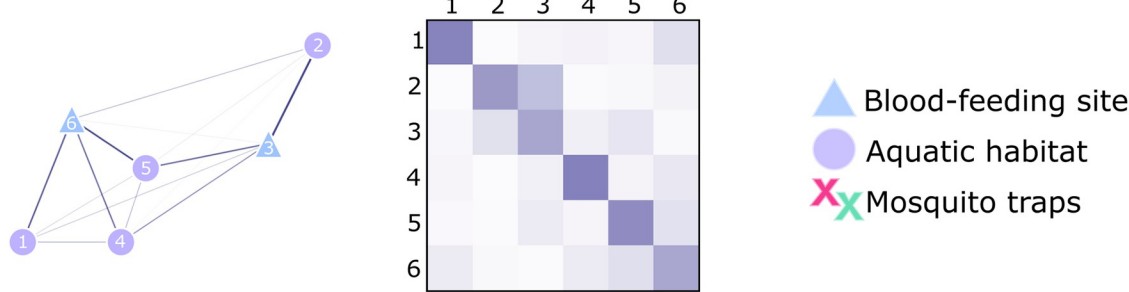

## C. Heterogeneous site types with traps

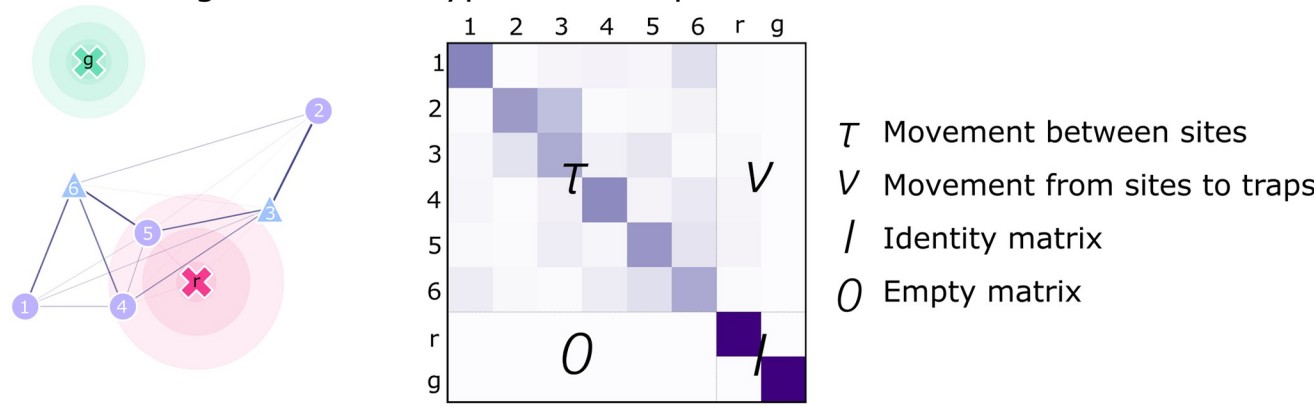

**Fig 2. Demonstration of landscape specification in MGSurvE.** (A) A metapopulation consisting of six mosquito population nodes is depicted on the left, with a corresponding movement matrix depicted on the right (shades represent daily per-capita movement probabilities). Here, movement probabilities depend only on distance, and are derived from a zero-inflated exponential dispersal kernel with a staying probability of 0.75 and a mean dispersal distance, conditional upon movement, of 1. (B) The same metapopulation is depicted with resources attributed to nodes on the left (triangles represent blood-feeding sites, and circles represent water/egg-laying sites). The corresponding movement matrix is depicted on the right. A masking matrix is used to modify movement probabilities according to the resource type of the mosquito's current node (e.g., to account for the fact that a recently blood-fed mosquito is more likely to seek a node with water for egg-laying, etc.). (C) Two traps with maximum attractiveness (i.e., the probability a mosquito falls into a trap in its immediate vicinity) of 0.5 (red) and 0.3 (green), are incorporated into the metapopulation with resources attributed depicted on the left. The coordinates and attractiveness profiles of the traps are used to modify the movement matrix, depicted on the right. Here, $\tau$ is a square matrix in which each entry stores the probability of movement from site to site, $\nu$ contains the probabilities of flying from sites to traps, $I$ represents the identity matrix, and 0 represents a matrix of zeros. The structure of the additional rows and columns reflects the fact that traps are absorbing.

[25], and the movement matrices generated can be exported to disk and mosquito spatial simulation frameworks such as MGDrivE [16], MGDrivE 2 [22] and MBITES [22].

## Results

To demonstrate how the MGSurvE framework can be used to distribute traps on a landscape in order to minimize time to detection of an allele of interest, we compute and visualize optimal trap placement for two example species and landscapes: i) an *Ae. aegypti* population in the suburban landscape of Yorkeys Knob in Queensland, Australia, and ii) an *An. coluzzii* population on the island of São Tomé, São Tomé and Príncipe. Code for these examples is available at the MGSurvE repository (https://github.com/Chipdelmal/MGSurvE).

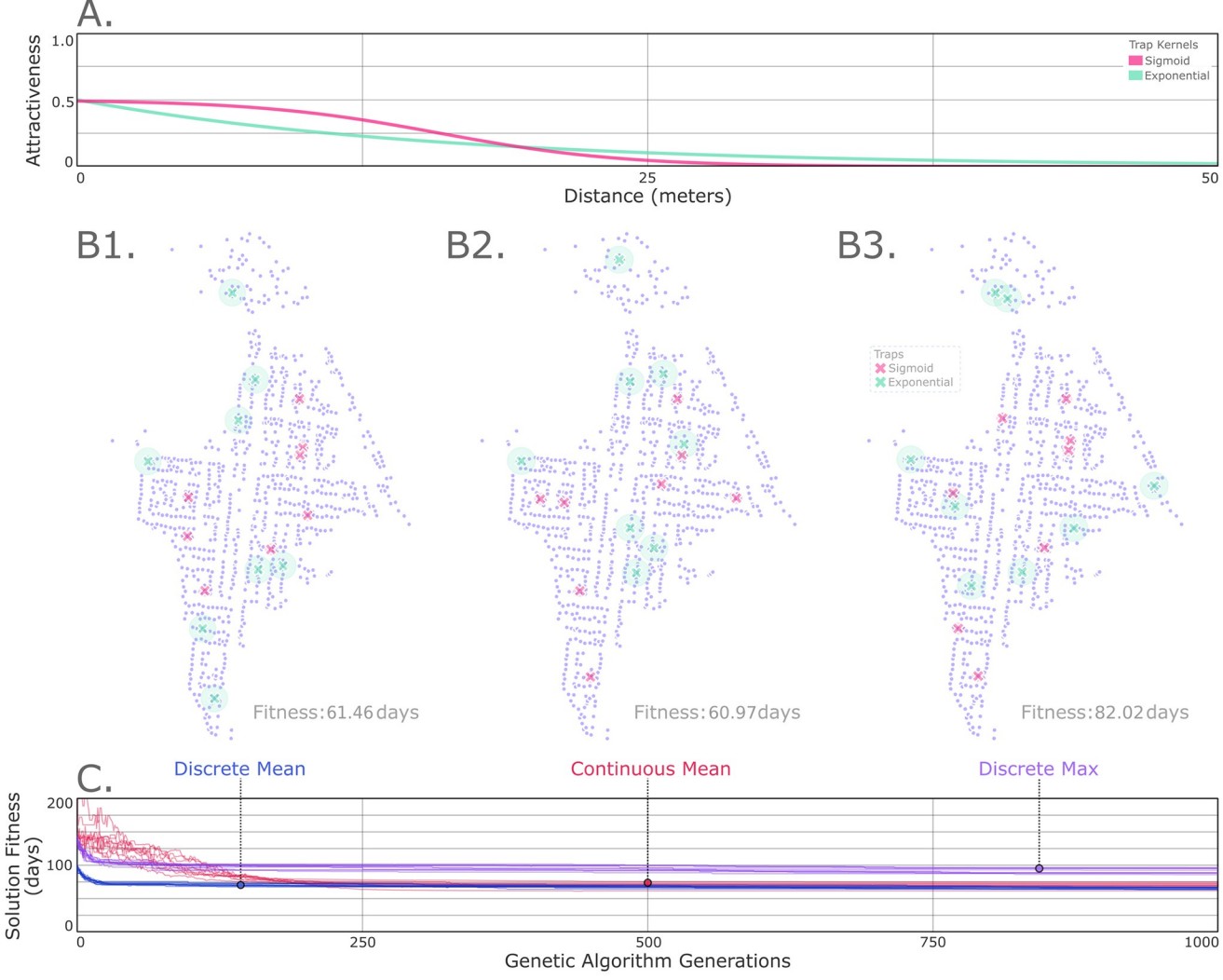

**Fig 3. Example optimal trap placement on a suburban landscape (*Ae. aegypti* in Queensland, Australia).** (A) We consider two types of traps with distinct attractiveness profiles represented by: i) an exponential kernel (green) with maximum attractiveness of 0.5 and a mean radius of attractiveness of 15.88 m, and ii) a sigmoidal kernel (magenta) with maximum attractiveness of 0.5, an inflection radius f 16 m and a shape parameter of 0.25. Household-based mosquito population nodes are depicted for Yorkeys Knob (B). Daily movement probabilities between households are derived from a zero-inflated exponential kernel with a daily staying probability of 72% [19, 27], and a mean dispersal distance conditional upon movement of 54 m [20]. The genetic algorithm of MGSurvE distributes traps such that the mean (B1–2) and maximum (B3) expected times for a given mosquito to be trapped, considering all possible origin sites on the landscape, is minimized. The base layer for map including buildings footprints can be downloaded from OpenStreetMap (https://www.openstreetmap.org/search?query=yorkeys%20knob#map=16/-16.8125/145.7273).

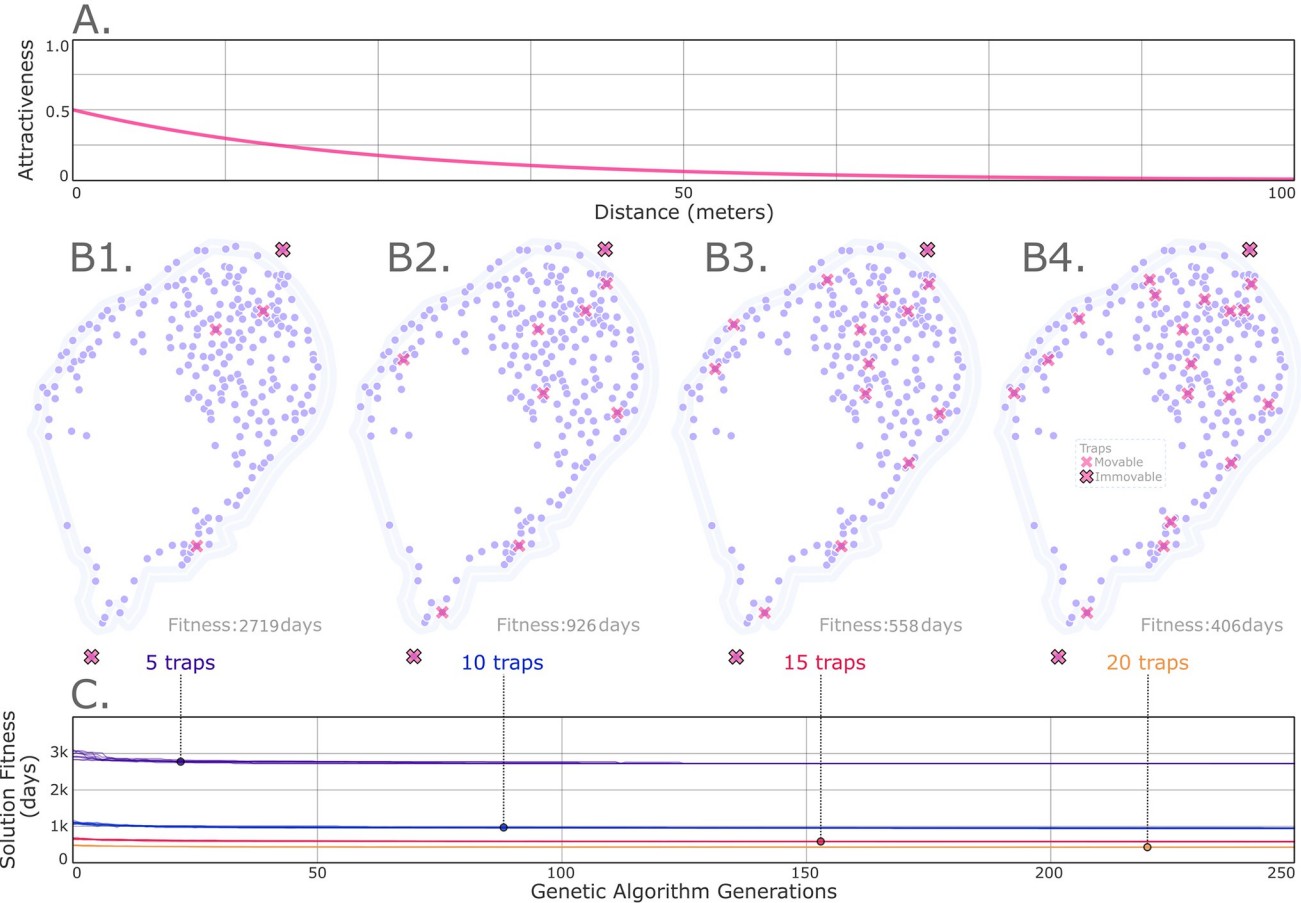

**Fig 4. Example optimal tap placement on an island landscape (*An. coluzzii* in São Tomé, São Tomé and Príncipe).** Placements of an increasing number of traps with exponentially-decaying attractiveness kernels (A) are optimized (5, 10, 15 and 20 traps). Two of these traps are fixed in position for each demonstration (black-edged crosses). Mosquito population nodes are depicted for São Tomé (B); which represent villages and suburbs of comparable size, aggregated to maintain a minimum distance of 500 m between nodes. Daily movement probabilities between localities are derived using an ecology-motivated algorithm through a resistance landscape calibrated to a daily staying probability higher than 0.97 and mean lifetime dispersal distance of 7.0 km [21, 32, 33]. The genetic algorithm of MGSurvE distributes traps such that the mean expected time for a given mosquito to be trapped, considering all possible origin sites on the landscape, is minimized (C). Here, each trace representing an independent run of the full optimization cycle. We consider optimally placing 5 (B1), 10 (B2), 15 (B3) and 20 (B4) traps on the landscape, each having an exponential kernel with a mean radius of attractiveness of 24 m [28] and maximum attractiveness of 0.5. The island's shapefile used in this manuscript is public domain and can be downloaded from Natural Earth (http://www.naturalearthdata.com/download/10m/cultural/ne_10m_admin_0_sovereignty.zip).

## Discrete and continuous optimization on a suburban landscape (*Ae. aegypti* in Queensland, Australia)

Here, we demonstrate the application of MGSurvE to distribute traps in order to minimize time to detection of an allele of interest for *Ae. aegypti* populations in the suburb of Yorkeys Knob 17 km northwest of Cairns, Queensland, Australia. Yorkeys Knob was a trial site for a successful release of *Wolbachia*-infected mosquitoes in 2011 [26] (Fig 3B). *Ae. aegypti* is a relatively local disperser, and so households serve as an appropriate population node for the landscape. Resource types do not need to be specified, as in this demonstration, we are assuming households provide all the feeding and breeding resources required by *Ae. aegypti*. This assumption, however, can be relaxed if we were interested in studying the effects of resource heterogeneity in the way we should approach our surveillance task (for tutorials on how to

incorporate these factors, see the online documentation at https://chipdelmal.github.io/MGSurvE/build/html/demos.html, and the model description in S1 Text). Household coordinates in Yorkeys Knob were sourced from OpenStreetMap (https://www.openstreetmap.org/), and daily movement probabilities between households were derived from a zero-inflated exponential kernel with a daily staying probability of 72% [19, 27], and a mean dispersal distance conditional upon movement of 54 m [20].

In optimizing trap placement, we consider two types of traps with distinct attractiveness profiles represented by: i) an exponential kernel with a mean radius of attractiveness of 16 m [28], and ii) a sigmoidal kernel with an inflection radius of 16 m and shape parameter of 0.25 (Fig 3A). Both traps have maximum attractiveness of 0.5, meaning that, for a trap placed within a population node, the ratio of mosquitoes that enter the trap to those that do not per time-step is 0.5:1 (i.e., in this case, half of mosquitoes in the same node as the trap enter the trap per day). In Yorkeys Knob, 16 traps were assigned, to reflect the number of traps used to monitor the *Wolbachia* trial at this site [26]. Half were assigned the exponential kernel, and half were assigned the sigmoidal kernel, to demonstrate the simultaneous placement of two trap types. We considered both discrete and continuous optimization cases to compare their results, although using the discrete case is preferred due to the fact that traps would likely be assigned to households without a more precise location being specified. The placement of each trap was then optimized according to two fitness functions—minimizing the mean and maximum expected times for a given mosquito to be trapped, considering all possible origin sites on the landscape. The GA implemented default mutation and crossover operators with 10 stochastic repetitions and 2,000 generations of the optimization process for each scenario (traces in Fig 3C). The code for this analysis is included in the S2 Text and at https://github.com/Chipdelmal/MGSurvE/tree/main/MGSurvE/demos/YKN, with additional video commentary and explanations available at https://youtu.be/RhYmeJ3XZ_8 (minutes 28 to 30). The operators and selected parameters for the algorithm are described in S1 Text. The resulting trap distributions are depicted in Fig 3B1 and 3B2 (for the case of the mean expected time for a mosquito to be trapped) and Fig 3B3 (for the case of maximum expected time for a mosquito to be trapped). From this output, we see that the GA distributes traps throughout the landscape, often placing them within concentrations of nodes, and sometimes in regions where sub-networks connect to the main section of the landscape, all of which are consistent with faster trapping times. The discrete and continuous mean times converged to similar optimum trapping times (61.46 and 60.97 days, respectively), despite their drastically different computational optimization evolutionary processes (Fig 3C shows the continuous case slowly moving towards the optimum, whereas the discrete case moves most in the early phases of the process). The maximum discrete optimization fitness metric seeks to minimize the time it would take for a mosquito to fall into any trap, beginning from the trap for which this time is longest. As such, this approach tends to prioritize placing traps as far from each other as possible whilst covering the most remote locations (e.g., the eastern-most households in our landscape). In this case, the algorithm converged to a minimum trap time of 82 days, beginning from the trap for which this time is longest. This is an important metric in situations where confinement of a certain allele/trait is paramount.

## Discrete optimization on an island landscape (*An. coluzzii* in São Tomé, São Tomé and Príncipe)

In the second example, we use MGSurvE to distribute traps in order to minimize time to detection of an allele of interest for *An. coluzzii* populations on the island of São Tomé, São Tomé and Príncipe. São Tomé, an island 225 km west of the coast of Gabon, has been

identified as a suitable candidate for field trials of gene drive mosquitoes due to the presence of a single dominant malaria vector species (*An. coluzzii*), relative isolation from mainland Africa, and a history of ecological studies of mosquitoes on the island [29] (Fig 4B). *An. coluzzii* can disperse relatively large distances [20], and so villages and suburbs of comparable size serve as appropriate population nodes. Due to the scale of these localities, and the fact that they were aggregated in a meta-population scheme, resource types do not need to be specified here as each village contains all the resources needed for mosquitoes to flourish. Locality coordinates in São Tomé were sourced by aligning locations in the São Tomé and Príncipe census (https://projectsportal.afdb.org/dataportal/VProject/show/P-ST-KF0-001) with locations in Google Maps (https://www.google.com/maps), and the DBSCAN algorithm (Density-Based Spatial Clustering of Applications with Noise) [30] was used to aggregate nearby localities in order to maintain a minimum distance of 500 m between nodes. Daily movement probabilities between localities were derived using an ecology-motivated algorithm in which mosquito movement is simulated as correlated random walks through a resistance landscape, with resistance being provided by elevation and land use [31]. Data from mark-release-recapture experiments on *An. gambiae* sensu lato [21, 28, 32] were used to calibrate the movement model according to a daily staying probability higher than 0.97 and mean lifetime dispersal distance of 7.0 km.

In optimizing trap placement, this time we consider a variable number (5–20) of a single type of trap with an attractiveness profile represented by an exponential kernel with a mean radius of attractiveness of 24 m [28] and maximum attractiveness of 0.5 (Fig 4A). We used discrete optimization due to the spatial scale, at which traps are most likely to be placed within localities. The placement of each trap was optimized according to a fitness function corresponding to minimizing the mean expected time for a given mosquito to be trapped, considering all possible origins on the landscape. The GA again implemented default mutation and crossover operators, with the code for this analysis being available in S2 Text and at https://github.com/Chipdelmal/MGSurvE/tree/main/MGSurvE/demos/STP (the description of the selected parameters for the operators is available in S1 Text). The resulting trap distribution is depicted in Fig 4B1–4B4 for the cases of 5, 10, 15 and 20 traps, respectively. This demonstration is different from the last one as we are increasing the number of traps whilst keeping the optimization fitness goals the same, and we are adding two immovable traps (north and south of the island). We can see that the algorithm balances the most transited nodes (in the northern region) while seeking to cover some of the least-connected sections of the landscape (in the south) as we increase the number of available traps. This is a result of the fitness function representing the average maximum time to detection beginning anywhere on the landscape. Additionally, we can see in Fig 4B and 4C that doubling the number of traps from 5 to 10 has a drastic impact on reducing the mean time taken for mosquitoes to fall into traps (from 2,719 to 926 days). From this point, adding 10 more traps would only reduce the time to 406 days, which may inform cost-effectiveness considerations.

## Supporting information

**S1 Text. Description of the modeling framework.** A description of the mathematical equations that govern mosquito movement and trap attractiveness implemented as an absorbing Markov chain.
(PDF)

**S2 Text. Example use cases in suburban and island landscapes.** Code used to optimize trap placement for *Ae. aegypti* in Queensland, Australia and *An. coluzzii* in São Tomé, São Tomé

and Príncipe, as described in the manuscript.
(PDF)

## Acknowledgments

We thank Dr. Tomás León and Dr. Jared Bennett for steering the initial phase of package development, and for testing software applications. We also thank Elijah Bartolome, Topiltzin Hernández Mares, Lillian Weng, Xingli Yu, Ayden Salazar, and Joanna Yoo for help with developing and testing the package.

## Author Contributions

**Conceptualization:** Héctor M. Sánchez C., David L. Smith, John M. Marshall.

**Formal analysis:** Héctor M. Sánchez C., John M. Marshall.

**Funding acquisition:** John M. Marshall.

**Investigation:** Héctor M. Sánchez C.

**Methodology:** Héctor M. Sánchez C., David L. Smith, John M. Marshall.

**Project administration:** Héctor M. Sánchez C.

**Resources:** John M. Marshall.

**Software:** Héctor M. Sánchez C.

**Supervision:** David L. Smith, John M. Marshall.

**Validation:** Héctor M. Sánchez C., John M. Marshall.

**Visualization:** Héctor M. Sánchez C.

**Writing – original draft:** Héctor M. Sánchez C., John M. Marshall.

**Writing – review & editing:** Héctor M. Sánchez C., David L. Smith, John M. Marshall.

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
