## [Decision Letter · Decision Letter 0]

23 Aug 2023

Dear PhD Sanchez C.,

Thank you very much for submitting your manuscript "MGSurvE: A framework to optimize trap placement for genetic surveillance of mosquito population" for consideration at PLOS Computational Biology.

As with all papers reviewed by the journal, your manuscript was reviewed by members of the editorial board and by several independent reviewers. In light of the reviews (below this email), we would like to invite the resubmission of a significantly-revised version that takes into account the reviewers' comments.

We cannot make any decision about publication until we have seen the revised manuscript and your response to the reviewers' comments. Your revised manuscript is also likely to be sent to reviewers for further evaluation.

Sincerely,

Alex Perkins

Academic Editor

PLOS Computational Biology

Feilim Mac Gabhann

Editor-in-Chief

PLOS Computational Biology

Reviewer's Responses to Questions

**Comments to the Authors:**

Reviewer #1: This manuscript describes a new software package that the authors have developed to assist researchers in planning field experiments of genetics-based mosquito control strategies. The software runs an algorithm to optimise trap placement for a given scenario, based on inputs including trap type and point locations relevant to the mosquito population of interest (e.g. house locations for the urban dwelling, short dispersing Aedes aegypti, village locations for longer dispersing Anopheles gambiae). The primary motivation is to support field testing of mosquitoes using gene drive technology to reduce the potential of the target population to transmit disease. This is a somewhat niche application - no gene drive mosquito has yet been field-tested, though several candidates for field testing are in active development - though trap placement and population monitoring more generally is a very important consideration for those working in this area. This is a potentially very useful tool for the audience it is aimed at, though we have a number of comments and suggestions that we feel might improve its clarity and effectiveness. These comments are based on both reading the manuscript and our experience of installing and running the worked examples.

General comments

• It could be more clearly stated that this tool is to optimise the monitoring of a closed population (e.g. that on an island). A version that is able to optimise trap placement to monitor the spread of a transgene in an open population would be a valuable extension (though perhaps out of scope at this time).

• For one or both of the worked examples, it would have been particularly useful to have a results section with analysis of how much is gained from each additional trap. e.g. number of traps on the x-axis, and time to observe y fraction of mosquitoes on the y axis.. This kind of analysis is something a field-test planner might want to do, so showing how the package can be used this way would help to illustrate its usefulness.

Medium comments

Clarity of mosquito model description

• We have assumed from the text that the mosquito model acts on a discrete timescale with daily increments? Is this right? Please make this explicit (apologies if we missed it!).

• Are we right to understand that the probability that a mosquito disperses from a to b is dependent on the density of other sites in the vicinity of the focal site (because the probabilities are normalised by their sum)? What you are describing are thus *relative* movement probabilities? Please clarify in the main text (lines 79-96).

• Are we right to understand there is no assumption on population sizes – they are implicitly assumed to be equal? If some nodes have larger populations than others, will this affect the results? (I suppose not if it is assumed that the probability of dispersing is not dependent on population size?).

Clarity of algorithm description. The genetic algorithm used to optimise trap placement needs to be explained in a bit more detail and clarity for the benefit of those of us not versed in genetic algorithm theory. In non-specialist language, what does the algorithm actually do?

• A number of words from GA theory are used without clear explanation of what they mean in this specific context. In particular: “Chromosomes”, “mutation”, “selection”, “crossover”. Please elaborate on these, taking care to avoid confusion given the biological nature of the application!

• For a given arrangement of traps, how is the fitness function calculated? Is it deterministic or estimated from stochastic simulations?

• How is convergence of the algorithm evaluated/monitored? What is a typical number of iterations needed? Does it give the same (or very similar) outputs on each run?

• How efficient is it - how many sites could it handle before becoming unwieldy?

Package installation/testing

• We installed the package on a windows 10 x64-based PC. This was successful when following the “Uneventful instructions”, after installing cartopy and libpysal additionally – we felt the documentation could be a bit clearer with respect to installing these additional packages.

• We tried running the examples from the paper. At first we had a little trouble before learning that we needed to download the csv data point files – the README in the github demo folder could be clearer about the need to download the csv files if not running from bash.

• YKN discrete takes very long to run, so we did not run it all! We had more success with the ‘Quickstart tutorial’, which took a few minutes.

Minor comments

• Fig. 1. Where it is first mentioned in the text, it could be clearer that the figure describes the structure of the method and not the visualisation itself.

• Fig. 3. The figure legend describes the plots in a different order to the figure itself! (B,C,A).

• Fig. 3. C needs more explanation - the two axes need some elaboration. NB 'generations' and 'fitness' obviously have resonance to both both the mosquito biology and GA terminology, so there is potential for confusion. Also the x-axis seems way over-stretched, only really interesting in the lower range.

• Sao Tome example. A daily (? or generational?) staying probability of 0.05 of staying in a village seems very low!!?

• Line 227. I don’t think ‘amplitude’ is a conventional word for describing this kind of dispersal kernel, and unclear what it means..?

• Fig. 2 could benefit from a colour scale bar? And a legend for the map symbols. On the matrix (C), spell out r and g (red and green, presumably).

• Figs 3 and 4. Put the plot labels A and C in the top left corner, hard to notice them where they are!

• What are the little numbers at the bottom right of each map plot?

• We didn’t find the supporting videos very instructive at all. They are very slow and with no narration it is hard to see what is going on. Perhaps they could show how the inputs are used, and include some narration of what is going on?

Reviewer #2: The manuscript presents MGSurvE, a framework to investigate optimal trap placement. Optimal trap placement is an essential task for many applications of genetic monitoring. The manuscript was mostly easy to read for people with a background in genetic algorithms and mosquito dispersal. But sometimes the manuscript felt repetitive to read and incomplete. Some sentences are too complex and challenging to read, e.g. lines 3-6 or Fig 2 lines 2-5. Split them up into multiple sentences for an easier read.

1. Line 11-12; what is split dive? Can you explain in a sentence how it works. How is it different from gene-drive.

2. Line 71-72; how much is local dispersal and greater distance? Can you provide an approximate distance from the reference [19] and [20] in the text

3. Line 73-77; You claim that MGSurvE can represent specific resources e.g. blood, sugar, and water, but none of your examples in the manuscript uses the biological modifying parameters. There is also no definition in the “S1 Text: Model description” of how the “biological modifying parameters” or “sex-specific” modify the movement.

4. Line 88; a figure in the supplement showing the different types of kernels of mosquito movement or trap attractiveness would be helpful for readers with a biological background.

5. Line 111-144; The provided explanation of the genetic algorithm (GA) is only understandable for people who know how GA works and even then it is hard to understand. How are the traps on the chromosome as allele optimized? Can you provide a figure in the supplement for easier understanding?

6. Line 171; the reference [28] for local disperser differs from the reference [19] mentioned on line 71. Why do you have two different references at two different locations in the manuscript? Should the reference be cited together if they both report the local dispersal of A. aegypti? Again can you provide an approximate distance from the reference?

7. Line 191-202; I missed a summary of the optimization results in the text. No information about the optimization parameter in the main text is provided e.g. how many generations and replicates were used to get the results. How many traps and why that many traps? What was the mean and max time expected time for a given mosquito to be trapped? Why are the references to the videos not mentioned in the mean text? It feels odd that the Fig 3 text provides more results information than the main text.

8. Figure 3; It would be nice to have a figure legend in the plots. Figure labels A, B1-3, and C are hidden in the figure and should be located outside of the figure in black. What is the meaning of the colors in Figure 3C? What is the meaning of the grey number in Figure 3B? What was the optimal number of traps? Move the reference to the videos to the main text as it does not describe anything in the figure.

9. Line 212; Why does the resource type not needed to be specified, explain more clearly what the scale of the localities means, e.g. due to the 500m minimum distance between nodes all the resources are provided within each node.

10. Line 205-207; Why is more information about the study side e.g. ‘west of the cost Gabon’ provided in the Figure 4 text than in the main text?

11. Line 228-240; I missed a summary of the optimization results in the text. What was the optimal number of traps? Also same issues as in point 7: No information about the optimization parameter in the main text are provided e.g. how many generations and replicates were used to get the results. What was the mean and max time expected time for a given mosquito to be trapped for the different number of traps. Why are the references to the videos not mentioned in the mean text?

12. Figure 4; The two immovable trap is not described in the figure text. It would be nice to have a figure legend in the plots. Figure label A, B1-3, and C are hidden in the figure and should be located outside of the figure in black. What is the meaning of the colors in Figure 4C? What is the meaning of the grey number in Figure 4B? Move the reference to the videos to the main text.

13. What type of application do you need to specify different types of resources? Why is there no example of different types of resources? One of the examples could have been modified to use the different resource types. It would have been interesting to see the difference in the optimal trap placement.

14. I would have liked to see an overview of different used kernel definitions for traps (mentioned in line 100) and mosquito dispersal in the supplement.

15. Line 261-262; Could you elaborate in the text on what would change when taking mosquito life history and intergenerational movement into account? Would only the time change or also the optimal trap placement? Is it really necessary to include the mosquito life history for optimal trap placement?

16. Line 242; Include the DOI (10.5281/zenodo.8087603) of the software version used in the manuscript

17. What is the unit of the expected number of time steps, e.g. days, weeks, generations?

Typos:

* ‘S’ missing In author summary without line numbering approx on line 4: are considered aS novel tool

S1 Text: Model description

* The online documentation is much more appealing than the model description. The model description would benefit if the information in the “Pkg Breakdown” section were included alongside the formulas in the model description.

* How does the biological modifying parameter modify the shape parameter ?

* Multiple parameters are not defined in the text or do not appear in the formulas, e.g. tj, tj^, ^trap, , , …

* Typo: remove ‘g’ in “Markov chainG properties”

S2 Text: Example use case

* For x0 and b use the distance in meters for easier understanding, e.g. 1/16 = 0.0629, or define from where the value 0.0629534 originates.

* I miss the definition of global parameters, e.g.TRPS_NUM and GENS

* What is the difference between optimizeTrapsGA and optimizeDiscreteTrapsGA. I assume continuous and discrete optimization. Clarify that in the “Yorkey’s Knob” example.

* Many parameters are not described, e.g. pop_size=’auto’, …

* Examples contain errors in the trap type specification of Sao Tome in tKer params ‘b’.

* What does the function deepcopy?

Reviewer #3: This manuscript introduces and describes MGSurvE, a software developed to optimize placement of ovitraps to detect mosquito populations with the goal of early detection of alleles, either those that are introduced or those that may be resistant or antagonistic to introduced genes. The manuscript is a well-written description of the software, and the authors present results for two scenarios in particular for which the software may be useful. The software is clearly important to trap placement and design of experiments that involve trap placement. I only have a few comments and questions about the manuscript in its current form.

1. The authors mention integration of MGSurvE with other software developed by members of this group. In particular, the authors write “MGSurvE may be used in parallel with MGDrivE or MGDriv

---

## [Decision Letter · Decision Letter 1]

2 Apr 2024

Dear PhD Sanchez C.,

We are pleased to inform you that your manuscript 'MGSurvE: A framework to optimize trap placement for genetic surveillance of mosquito populations' has been provisionally accepted for publication in PLOS Computational Biology.

Best regards,

Feilim Mac Gabhann, Ph.D.

Editor-in-Chief

PLOS Computational Biology

Reviewer's Responses to Questions

**Comments to the Authors:**

Reviewer #2: The manuscript presents MGSurvE, a framework to investigate optimal trap placement. Optimal trap placement is an essential task for many applications of genetic monitoring. The authors addressed all my comments from the first review. The clarity of the text is greatly improved in the revised manuscript.

I have only few minor comments of two potential typos

Line 63: Point at end of sentence is missing between “visualized These”

Fig 2: Missing “o” at “ radius Of 16 m”

Reviewer #3: I have no additional comments for the authors. I think the manuscript is excellent in its current form.

**Have the authors made all data and (if applicable) computational code underlying the findings in their manuscript fully available?**

Reviewer #2: Yes

Reviewer #3: Yes

PLOS authors have the option to publish the peer review history of their article (what does this mean?). If published, this will include your full peer review and any attached files.

Reviewer #2: No

Reviewer #3: No

---

## [Editor Report · Acceptance letter]

30 Apr 2024

PCOMPBIOL-D-23-01021R1 

MGSurvE: A framework to optimize trap placement for genetic surveillance of mosquito populations

Dear Dr Sánchez C.,

I am pleased to inform you that your manuscript has been formally accepted for publication in PLOS Computational Biology. Your manuscript is now with our production department and you will be notified of the publication date in due course.

With kind regards,

Judit Kozma
